# The Adaptive Complexity of Maximizing a Gross Substitutes Valuation

**Ron Kupfer**
The Hebrew University of Jerudalem
ron.kupfer@mail.huji.ac.il

**Sharon Qian**
Harvard University
sharonqian@g.harvard.edu

**Eric Balkanski**
Harvard University
ericbalkanski@g.harvard.edu

**Yaron Singer**
Harvard University
yaron@seas.harvard.edu

## Abstract

In this paper, we study the adaptive complexity of maximizing a monotone gross substitutes function under a cardinality constraint. Our main result is an algorithm that achieves a $1 - \epsilon$ approximation in $\mathcal{O}(\log n)$ adaptive rounds for any constant $\epsilon > 0$, which is an exponential speedup in parallel running time compared to previously studied algorithms for gross substitutes functions. We show that the algorithmic results are tight in the sense that there is no algorithm that obtains a constant factor approximation in $\tilde{o}(\log n)$ rounds. Both the upper and lower bounds are under the assumption that queries are only on feasible sets (i.e., of size at most $k$). We also show that under a stronger model, where non-feasible queries are allowed, there is no non-adaptive algorithm that obtains an approximation better than $1/2 + \epsilon$. Both lower bounds extend to the class of OXS functions. Additionally, we conduct experiments on synthetic and real data sets to demonstrate the near-optimal performance and efficiency of the algorithm in practice.

## 1 Introduction

In this paper, we study the problem of maximizing gross substitutes functions in the adaptive complexity model. Gross substitutes are an extremely well-studied class of functions in microeconomics. The concept of gross substitutes was first introduced in the seminal work by Arrow and Debreu as a sufficient condition on the valuation functions of buyers to guarantee the existence of equilibria in markets with indivisible items [1]. It was later shown to also be a necessary condition [20]. Gross substitutes functions are also studied in the contexts of stable matchings in two-sided markets [2, 33], combinatorial auctions [3], and trading networks [23, 25], and have been rediscovered in multiple fields under different names. We refer the reader to [31] for a survey of the different definitions.

In theoretical computer science and optimization, gross substitutes are considered as a subclass of *submodular functions*, as they satisfy the diminishing returns property. For monotone submodular functions, it is well known that a greedy algorithm that iteratively selects the element with the maximal marginal contribution to its current solution obtains a $1 - 1/e$ approximation for maximization under a cardinality constraint [30] and that this bound is optimal for polynomial-time algorithms [29, 19]. For gross substitutes functions, the greedy algorithm returns an *optimal* solution [13]. Thus, from a purely algorithmic perspective, gross substitutes represent an important subclass of submodular functions: it is the most expressive class of submodular functions that can be optimized *exactly* under cardinality constraints in polynomial time. Not only is gross substitutability a sufficient condition for the optimality of greedy, but it is also a necessary condition [31].[1]

A recent line of work began investigating the *adaptive complexity* of submodular optimization [8, 6, 15, 11, 18, 9, 5, 17, 12, 7, 10, 16, 26]. The adaptive complexity model was introduced in [8] as an information theoretic measure for the parallel runtime of an algorithm. Informally, the adaptivity of an algorithm is its number of sequential rounds, when each round can perform polynomially-many function evaluations in parallel. Since the greedy algorithm adds a single element to the current solution at every iteration, it has adaptivity that is linear in the cardinality constraint $k$, which, in the worst case, is $\Omega(n)$. Until recently, there was no known constant factor approximation algorithm whose adaptivity is sublinear in $k$ for maximizing a submodular, or even a gross substitutes function.

The main result in [8] is an algorithm that obtains a constant factor approximation arbitrarily close to $1/3$ in $\mathcal{O}(\log n)$ rounds, which was an exponential speedup in parallel runtime for maximizing monotone submodular functions under a cardinality constraint. They also showed that there is no $\tilde{o}(\log n)$ adaptive algorithm that achieves a constant approximation. The algorithm in [8] uses a technique called adaptive sampling that was first extended by [6, 15] to obtain an approximation that is arbitrarily close to the optimal $1 - 1/e$, and then by other papers in this genre [18, 5, 17, 26].

Although there has been a great deal of work on submodular maximization in the adaptive complexity model, this work has focused on general submodular functions and little is known about the adaptive complexity of maximizing gross substitutes functions. On one hand, gross substitutes are a superclass of additive and unit-demand functions. For additive and unit demand algorithms, it is trivial to obtain an approximation arbitrarily close to optimal (i.e. $1 - \epsilon$, for any constant $\epsilon$) with only 1 round. On the other hand, gross substitutes are a *subclass* of submodular functions. Thus, all the results in the adaptive complexity model for submodular functions also apply to gross substitutes and there is a $\mathcal{O}(\log n)$-adaptive algorithm that obtains an approximation arbitrarily close to $1 - 1/e$ for maximizing monotone gross substitutes under a cardinality constraint. But to obtain near optimal results, the only algorithm known is the greedy algorithm whose adaptivity is linear in $k$.

> *How many rounds are needed to find a (near) optimal solution to the problem of maximizing gross substitutes under a cardinality constraint?*

**Main results.** We first show that the number of rounds needed to find a solution that is arbitrarily close to optimal for maximizing monotone gross substitutes under a cardinality constraint is $\mathcal{O}(\log n)$. In particular, for any $\epsilon > 0$, there exists an $\mathcal{O}(\log(n)/\epsilon^3)$ adaptive algorithm that obtains a $1 - \epsilon$ approximation in expectation. This near-optimal algorithm provides an exponential improvement in parallel runtime compared to previous algorithms for maximizing gross substitutes. We also provide two lower bounds. The first shows that there is no non-adaptive, i.e. 1-adaptive, algorithm that obtains an approximation better than $1/2 + \epsilon$ for maximizing monotone gross substitutes under a cardinality constraint. This hardness result shows a sharp separation between gross substitutes and additive and unit demand functions which can both be optimized arbitrarily well in a single round. The second lower bound is a conditional lower bound. Assuming that the algorithm queries sets of size $\mathcal{O}(k)$, there is no $\tilde{o}(\log n)$ algorithm that obtains a constant approximation for maximizing monotone gross substitutes under a cardinality constraint.

Furthermore, we conduct experiments on synthetic bipartite graphs and Twitter data. We observe that the algorithm has near-optimal performance while running in exponentially fewer parallel rounds. Additionally, the adaptive algorithm outperforms its benchmarks on a range of different valuations. In practice, we observe that the true number of rounds the algorithm requires is much lower than the theoretical bound for the approximation guarantee.

## 1.1 Technical Overview

**The algorithm.** To show low adaptivity and near-optimal approximation, we first define two classes of algorithms called impatient greedy and stochastic greedy. Each iteration of a stochastic greedy algorithm selects an element whose marginal contribution is in expectation a $1 - \epsilon$ approximation to the optimal marginal contribution at that iteration. We show that this algorithm obtains a $1 - \epsilon$ approximation for gross substitutes.

Most low adaptivity algorithms for submodular functions use a technique called adaptive sampling [8, 15, 18, 17, 5, 6, 26]. Unfortunately, adaptive sampling does not guarantee that elements added to the solution have near-optimal contribution at each iteration, which fails to give near-optimal guarantees for gross substitutes (see example in Appendix A.2). Instead, the algorithm leverages a recent

adaptive sequencing technique [7] that guarantees near-optimality of the marginal contributions of elements added. However, adaptive sequencing from [7] has $\mathcal{O}(\log(n)\log(k))$ adaptivity.

To improve the adaptivity, we introduce the class of impatient greedy algorithms, which begin by adding elements as long as their marginal contributions are above some fixed threshold, i.e., not necessarily close to optimal. We show that with threshold $\frac{\text{OPT}}{\epsilon k}$, impatient greedy algorithms obtain near-optimal approximation guarantees for gross substitutes. The main algorithm employs the adaptive sequencing technique starting with a threshold equal to $\frac{\text{OPT}}{\epsilon k}$. With this low threshold, adaptivity is improved from $\mathcal{O}(\log(n)\log(k))$ to $\mathcal{O}(\log n)$ with an arbitrarily small loss in the approximation. In this paper, we only consider maximization under cardinality constraint and leave general matroid constraints for future work. Since the greedy algorithm does not yield the optimal solution for gross substitutes under matroid constraints, many of these techniques cannot be immediately extended.

**Lower bounds.** Previous lower bound constructions in the adaptive complexity model [8, 9] are submodular, but not gross substitutes, so novel constructions are needed. The functions we construct to be hard to optimize are OXS functions, thus our lower bounds also hold for this subclass of gross substitute functions. The main challenge in the analysis of $\tilde{o}(\log n)$ rounds lower bound is handling the subtle interactions between the different rounds of queries by the algorithm. Our approach is related to the round elimination technique from communication complexity.

## 2 Preliminaries

We assume value oracle access to a function $f : 2^N \to \mathbb{R}$. An algorithm is $r$-*adaptive* if it consists of $r$ sequential rounds where the algorithm may perform $\text{poly}(n)$ function evaluations $f(S)$ in parallel at every round. A function $f$ is *submodular* if it exhibits the diminishing returns property, i.e., $f_S(a) \geq f_T(a)$ for all $S \subseteq T \subseteq N$ and $a \in N \setminus T$, where $f_S(T) := f(S \cup T) - f(S)$ is the marginal contribution of $T$ to $S$. We abuse notation and write $f(a)$, $f_S(a)$ for $f(\{a\})$ and $f_S(\{a\})$ when clear from context. As discussed in the introduction, there exist many equivalent definitions for gross substitutes (GS), see [31] for a detailed survey. We give the definition which we use for the analysis. A function $f$ is *gross substitutes* if it is submodular and for all $S, T \subseteq N$ and $a \in S$, $f(S) + f(T) \leq \max_{b \in T} \{f(S \setminus a) + f(T \cup a), f(S \cup b \setminus a) + f(T \cup a \setminus b)\}$. We show our lower bounds for gross substitutes by constructing families of OXS functions. A function $f$ with $N = \{a_1, \ldots, a_n\}$ is a unit-demand function if there are $n$ positive weights $w_1, \ldots, w_n \in \mathbb{R}_+$ s.t. $f(S) = \max_{a_i \in S} w_i$. A function $f$ is an *assignment function (OXS)* if $f$ is the convolution of $r$ unit-demand functions $u_1, \ldots, u_r$: $f(S) = \bigvee_{i \in [r]} u_i(S) := \max_{\cup_{i \in [r]} S_i = S} \sum_{i \in [r]} u_i(S_i)$ where sets $S_1, \ldots S_r$ are a partition of $S$. These three classes are related as follows [27]: $OXS \subsetneq GS \subsetneq SM$.

## 3 $\mathcal{O}(\log n)$ Rounds Suffice for Near Optimal Approximation

In this section we describe an algorithm for maximizing gross substitutes functions which has low adaptivity and returns a solution whose approximation guarantee is arbitrarily close to optimal. To prove these properties we first define two classes of algorithms – *impatient greedy* and *stochastic greedy* algorithms. We show that impatient greedy algorithms yield low adaptivity algorithms for gross substitutes functions and that stochastic greedy algorithms yield approximately optimal algorithms. We then define our main algorithm which is both an impatient greedy algorithm and a stochastic greedy algorithm and can, therefore, instantiate the guarantees for both to show that the algorithm is $\mathcal{O}(\log n)$ adaptive and achieves an approximation guarantee arbitrarily close to optimal.

### 3.1 IMPATIENT GREEDY Analysis

An impatient algorithm first collects items with marginal contribution to the current set above some input threshold $t$. In the second stage, the algorithm adds the remaining elements using the greedy algorithm. We show that by choosing the correct threshold $t$, this algorithm performs nearly optimally for gross substitutes functions. This choice of the threshold is the crucial step in showing a good approximation guarantee is achievable in few rounds. All proofs are deferred to Appendix B.

With threshold $t = \frac{\text{OPT}}{\epsilon k}$, we show that stochastic greedy performs near optimally for gross substitutes. This follows from the fact that the number of elements chosen in the first loop is bounded by $\epsilon k$ (Lemma 3). We defer the analogous result for submodular functions to Appendix B.3.

---

**Algorithm 1** IMPATIENT GREEDY

---

1: Input $f(\cdot)$, $k$, $t$
2: $S \leftarrow \emptyset$, $X \leftarrow N$
3: **while** $X \neq \emptyset$ and $|S| < k$ **do**
4:      $X \leftarrow \{a : f_S(a) \geq t\}$
5:      $S \leftarrow S \cup \{a_i\}$ where $a_i$ is chosen u.a.r. from $X$
6: **while** $|S| < k$ **do**
7:      $X \leftarrow \{a : f_S(a) = \max_x f_S(x)\}$
8:      $S \leftarrow S \cup \{a_i\}$ where $a_i$ is chosen arbitrarily from $X$
9: **return** $S$

---

**Theorem 1.** *Given a monotone gross substitutes function $f : 2^N \to \mathbb{R}$, IMPATIENT GREEDY with threshold $t = \frac{OPT}{\epsilon k}$ returns a set $S$ such that $f(S) \geq (1 - \epsilon)\,OPT$.*

### 3.2 STOCHASTIC GREEDY Analysis

In this section, we show that the *stochastic greedy* algorithm can guarantee a strong approximation to the optimal solution of gross substitutes functions. At each step $i$ of the algorithm, noise parameters are sampled from distribution $\mathcal{D}_i$ and an approximate maximal element is chosen (Algorithm 2). For submodular functions, stochastic greedy algorithms give approximations arbitrarily close to $1 - 1/e$ with high probability when $k$ is sufficiently large [22]. Noisy versions of the greedy algorithm on submodular functions are a vast area of research [32, 24, 21, 34, 22]. However, for gross substitutes functions, this has not been studied.

We prove results for gross substitutes functions, which may be of interest even outside the context of adaptive complexity. In particular, we show that for gross substitutes functions, an algorithm that selects an element which is, in expectation, an $\alpha$ approximation to the element with the largest marginal contribution at each iteration provides an $\alpha$ approximation to the optimal solution. This idea is crucial in proving our main result in Theorem 3.

---

**Algorithm 2** STOCHASTIC GREEDY

---

1: $S \leftarrow \emptyset$
2: **for** $i \in [k]$ **do**
3:      $(\xi_i, \zeta_i) \sim \mathcal{D}_i$
4:      $X = \{a : f_S(a) \geq \xi_i \max_x f_S(x) - \zeta_i\}$
5:      $S \leftarrow S \cup \{a_i\}$ where $a_i$ is chosen u.a.r. from $X$
6: **return** $S$

---

We first state the following lemma from [31] which will be useful for bounding the marginal contribution obtained at each step of the algorithm. All proofs are deferred to Appendix C.

**Lemma 1** ([31])**.** *Let $f$ be a gross substitutes function, then $\forall S, T \subseteq [n]$ with $|S| = |T|$ and $s \in S \setminus T$, we have $f(S) + f(T) \leq \max_{t \in T \setminus S} \{f(S \cup t \setminus s) + f(T \cup s \setminus t)\}$.*

We can use this to show that at any iteration, there exists an element with high marginal contribution to the current solution (Lemma 5) and obtain the following approximation guarantee.

**Theorem 2.** *Given a gross substitutes function $f : 2^N \to \mathbb{R}$, let $S$ be the set of size $k$ selected by STOCHASTIC GREEDY with $\hat{\xi} = \min_i \mathbb{E}[\xi_i]$ and $\hat{\zeta} = \sum_i \mathbb{E}[\zeta_i]$. Then $\mathbb{E}[f(S)] \geq \hat{\xi}\,OPT - \hat{\zeta}$.*

It now follows that the stochastic variant of the greedy algorithm gives a good approximation to the maximal value when $\hat{\xi} \approx 1$ and $\hat{\zeta} \approx 0$. In the case where the expected noise is bounded and $\mathbb{E}[\xi_i] = 1 - \epsilon$ and $\zeta_i = 0$ for all $i$, we can get a good approximation to the optimal solution in expectation, i.e. $\mathbb{E}[f(S)] \geq (1 - \epsilon)\,OPT$.

This is indeed the case for the algorithm discussed in the next section. We note that for submodular functions, while the approximation ratio of $1 - 1/e$ is preserved under noise, it is not always true that the noisy output is close to the output of the greedy algorithm. See example in Appendix A.1.

### 3.3 The Low Adaptivity Algorithm for Gross Substitutes

We now describe an algorithm for maximizing gross substitutes functions, GROSS SUBSTITUTES ADAPTIVE SEQUENCING (GSAS), which has $\mathcal{O}(\log n)$ rounds and returns a solution whose approximation guarantee is arbitrarily close to optimal. In the analysis, we show how to exploit the guarantees of the two variants presented previously by constructing an algorithm with similar approximation guarantees using a small number of adaptive query rounds. We analyze the performance for both submodular and gross substitutes monotone functions and show that for gross substitutes functions a $1 - \mathcal{O}(\epsilon)$ approximation is obtained[2]. All proofs are deferred to Appendix D.

**Adaptive sequencing.**  In this paper we develop an algorithm that is based on the *adaptive sequencing technique* recently proposed in [7] which was developed to obtain constant factor approximation guarantees under matroid constraints. The overwhelming majority of low-adaptivity algorithms use a different technique called *adaptive sampling* [8, 15, 18, 17, 5, 6, 26], which was introduced in [8]. Adaptive sampling algorithms sample a large number of sets of elements at every iteration to estimate marginal contributions. These estimates, which rely on concentration arguments, are then used to either add a random set $R$ to $S$ or discard elements with low expected contribution to $R \cup S$. Since gross substitutes functions are submodular, adaptive sampling provides a $1 - 1/e$ approximation but fails to give near-optimal guarantees (see Appendix A.2 for an example).

In contrast to adaptive sampling, adaptive sequencing techniques generate at every iteration a *single* random sequence $(a_1, \ldots, a_{|X|})$ of the elements $X$ not yet discarded. Let $A_l = (a_1, \ldots, a_l)$ be a sequence of elements. A prefix $A_{i^\star} = (a_1, \ldots, a_{i^\star})$ of the sequence is then added to the solution $S$, where $i^\star$ is the largest position $i$ such that a large fraction of the elements in $X$ has high contribution to $S \cup A_{i-1}$. Elements with low contribution to the new solution $S$ are then discarded from $X$.

---

**Algorithm 3** GROSS SUBSTITUTES ADAPTIVE SEQUENCING (GSAS)

1: Input $f(\cdot)$, $\epsilon$, $\Delta$, $t^\star$
2: $S \leftarrow \emptyset, t \leftarrow t^\star$
3: **for** $\Delta$ iterations **do**
4:      $X \leftarrow N$
5:      **while** $X \neq \emptyset$ **do**
6:          $a_1, ..., a_{k^*} \leftarrow$ random sampling from $X$ of size $k^* = \min(k - |S|, |X|)$
7:          $X_i \leftarrow \left\{ a \in X : f_{S \cup \{a_1, \ldots, a_{i-1}\}}(a) \geq t \right\}$ for all $i \in [k^*]$
8:          $i^* \leftarrow \min \left\{ i : |X_i| \leq (1 - \epsilon)|X| \right\}$
9:          $S \leftarrow S \cup \{a_1, ..., a_{i^*-1}\}$
10:          $X \leftarrow X_{i^*}$
11:      $t \leftarrow (1 - \epsilon)t$
12: **return** $S$

---

**Variants of Greedy.**  Adaptive sequencing described in [7] can be viewed as a parallel STOCHASTIC GREEDY algorithm. It starts with a high threshold $t^\star$ and lowers the threshold as the algorithm continues. The initial threshold $t^\star$ is set to $\max f(a)$ so that the algorithm will not discard good elements. In the case where $\mathtt{OPT} = \max f(a)$, the number of threshold decrements to guarantee a good approximation is $\mathcal{O}(\log(k))$, which results in the total adaptive complexity of $\mathcal{O}(\log(k)\log(n))$. Using the abstraction of IMPATIENT GREEDY, we can set a lower initial threshold $t^\star = \frac{\mathtt{OPT}}{\epsilon k}$ in GSAS to reduce complexity. We show that the number of rounds needed by the outer loop will decrease to $\Delta = \mathcal{O}(1/\epsilon^2)$. From Section 3.1, this threshold adjustment does not greatly effect the performance.

Note that GSAS requires the value of $\mathtt{OPT}$ to set the initial threshold. We can bypass this by running several estimations for $\mathtt{OPT}$ in parallel, setting $\mathtt{OPT} \in \left\{(1 + \epsilon)^i \max f(a) \mid i \in \left[\frac{\ln n}{\epsilon}\right]\right\}$, where the running time of each will be truncated by $\Delta$ iterations. We will show that a close approximation of $\mathtt{OPT}$ is sufficient. From now on, we denote the initial value of $t$ as $t^\star$ where $\frac{\mathtt{OPT}}{(1+\epsilon)\epsilon k} \leq t^\star \leq \frac{\mathtt{OPT}}{\epsilon k}$.

We now show our main results that GSAS achieves a $1 - \mathcal{O}(\epsilon)$ approximation in $\mathcal{O}(\log n)$ adaptive rounds. We start by showing logarithmic adaptivity. At a high level, low adaptivity follows from the fact that each inner iteration makes at most $\log(n)/\epsilon$ rounds and the outer loop runs $\Delta$ times.

**Lemma 2.** *Given $\epsilon > 0$ and $\Delta = 1/\epsilon^2$,* GSAS *terminates after $\mathcal{O}(\log(n)/\epsilon^3)$ rounds.*

We now outline a proof sketch for gross substitutes maximization with low adaptivity. We defer the analogous result for submodular functions and its proof to Appendix D.4.

**Theorem 3.** *For any monotone gross substitutes function $f$ and $\epsilon > 0$,* GSAS *is a $\mathcal{O}(\log(n)/\epsilon^3)$ adaptive algorithm that returns a set $S$ such that $\mathbb{E}[f(S)] \geq (1 - \mathcal{O}(\epsilon))\mathtt{OPT}$.*

*Proof Sketch.* Since the initial threshold of GSAS is lowered, we handle the first iteration separately and similarly to how we handle IMPATIENT GREEDY. We then show that remaining iterations behave similarly to STOCHASTIC GREEDY. Let $S_1$ be the set at the end of the first iteration. Then, either $S_1$ is optimal or is small in size w.h.p., i.e. $f(S_1) = \mathtt{OPT}$ or $|S_1| < 3\epsilon k$ (Lemma 6).

We now consider all remaining iterations and show that GSAS approximately maximizes a surrogate function $g(A) := f(S_1 \cup A)$ over cardinality constraint $k - |S_1|$. After the first iteration, there are no elements with marginal contribution exceeding $\frac{\mathtt{OPT}}{\epsilon k}$ and the algorithm can be reduced to the original version in [7] on $g$ after $S_1$ has been selected. The approximation guarantee follows from the fact that for each iteration, the threshold $t$ is an approximate upper bound on the maximal marginal contribution of an element $a$ to the intermediate solution $S$, i.e. $t \geq (1 - \epsilon) \max_a g_S(a)$ (Lemma 7). It then follows that GSAS behaves similarly to STOCHASTIC GREEDY where $\mathbb{E}[\xi_i] \geq 1 - 2\epsilon$ and $\zeta_i = 0$ (Lemma 8). We can then use Theorem 2 from STOCHASTIC GREEDY to show that indeed $S$ approximately maximizes $g$ with constraint $k - |S_1|$.

Combining this result with the analysis of the first iteration, we show that $S$ combined with $S_1$ approximately maximizes $f$ with constraint $k$. Finally, we handle the possibility of early termination. Since we miss at most $k$ elements, we have a loss of at most $tk = \epsilon\mathtt{OPT}$. Thus, we get that GSAS gives a $1 - \mathcal{O}(\epsilon)$ approximation. $\qquad\square$

The approximation can also hold with high probability using Markov's Inequality by running polynomially many copies of the algorithm and choosing the maximal one.

# 4 Lower Bounds

In this section, we present lower bounds on the adaptive complexity of maximizing gross substitutes. We first show that there is no 1-adaptive algorithm that obtains a $1/2 + \epsilon$ approximation, for any constant $\epsilon > 0$ (constructions and proofs deferred to Appendix E). This lower bound shows a sharp separation between gross substitutes and additive and unit demand functions, which can be optimized to be arbitrarily close to 1 in just one round.

**Theorem 4.** *There is no non-adaptive, i.e. 1-adaptive, algorithm that obtains, with probability $\omega(\frac{1}{n})$, a $1/2 + \epsilon$ approximation for maximizing monotone gross substitutes functions under a cardinality constraint, for any constant $\epsilon > 0$.*

We now show a lower bound for algorithms with multiple adaptive rounds. More precisely, we show that there is no $\tilde{o}(\log n)$ adaptive algorithm that obtains a constant approximation for maximizing OXS functions when the queries are of size $\mathcal{O}(k)$. The main challenge in extending the result from one round is handling the subtle interactions between the different rounds of queries.

We now discuss the assumption that the queries are of size $\mathcal{O}(k)$. We first note that in the context of learning or optimization from past observations and decisions, it is natural that past observations and decisions must also be feasible according to the problem constraint, i.e., of size at most $k$. In addition, we also note that most of the existing algorithms, e.g. greedy and local search, as well as the algorithm from the previous section, only query feasible sets of size at most $k$.

**Theorem 5.** *There is no $(\frac{\log n}{4 \log(\log n)} - 1)$-adaptive algorithm that obtains, with probability $\omega(\frac{1}{n})$, a $\frac{1}{\log n}$ approximation for maximizing monotone gross substitutes functions under a cardinality constraint when the queries are sets of size $\mathcal{O}(k)$.*

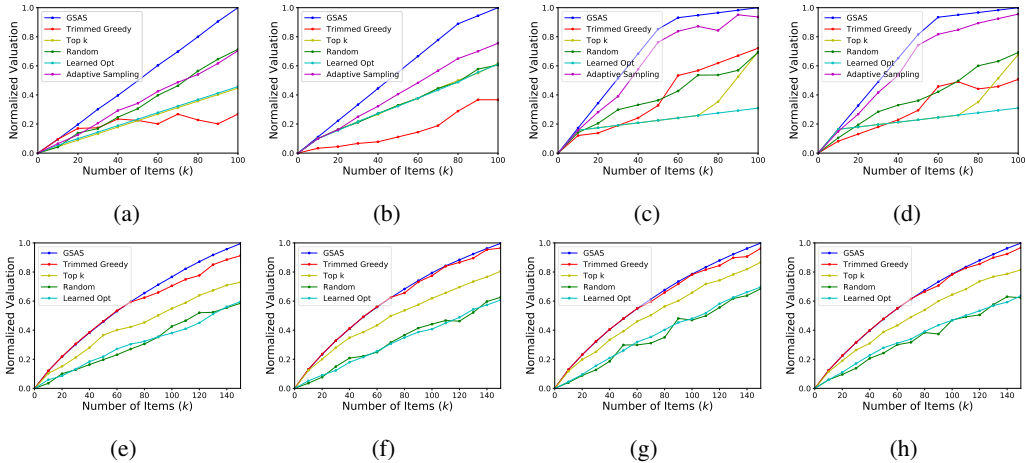

Figure 1: OXS valuation results on synthetic graphs G1, G2, G3 and G4 (Figures 1a, 1b, 1c, 1d) and tweets with #ad, #spon, #giveaway and #win hashtags G5, G6, G7 and G8 (Figure 1e, 1f, 1g, 1h).

## 5 Experiments

To evaluate the performance of GSAS, we conduct experiments on synthetic and real datasets.

**Experimental setup.** In our first set of experiments, we construct bipartite graphs with $n$ players, $m$ items (nodes) and valuations (edge weights) to simulate OXS valuations using synthetic and real data. We then compare the performance of GSAS against different benchmarks across varying values of $k$. We select $k = 100$ elements on synthetic data and $k = 150$ on real data. Our second experiment analyzes the spectrum of unit-demand and additivity. OXS valuations can be represented as the sum of max of item values. In one extreme, valuations are additive so that $v(A) = \sum_{a \in A} v(a)$; in the other extreme, they are unit-demand valuations so that $v(A) = \max_{a \in A} v(a)$. In this experiment, we explore the performance of GSAS by constructing OXS valuations that are strictly unit-demand, additive and in the spectrum and selecting $k = 32$ items for each valuation. For all of the experiments we have used $\epsilon = 0.1$.

**Benchmarks.** We compare GSAS to the following baselines. **Trimmed Greedy** is the GREEDY algorithm limited by the number of rounds used by GSAS to return a set of size smaller than $k$. Without this limitation, the algorithm returns the optimal value. **RANDOM** samples, in one round, $n$ many $k$-tuples and returns the best sample. **TOP-$k$** selects $k$ elements with largest marginal contribution to empty set in a single round. **Adaptive Sampling** adds sets of elements in each round by iteratively filtering out elements of low marginal contribution and selecting elements of high value [6]. **Learned Opt** first learns the OXS valuation function using sampling and then uses GREEDY to optimize the learned function [4]. We omit OPT from our plots since in our experiments OPT and GSAS are empirically indistinguishable.

### 5.1 Datasets

We briefly discuss the generation of synthetic graphs and the constructed Twitter network for the first set of experiments and OXS valuations for the second set. See Appendix F for more details.

**Synthetic graphs.** We generate the first graph by following the construction detailed in Appendix A.2 (G1) and a second using the construction detailed in Appendix E.3 with $n = 5000$ ground set elements (G2). We construct two additional random bipartite graphs with $n = 275$ ground set items and $m = 200$ players with the probability of an edge fixed at 0.25 (G3) and 0.75 (G4).

**Twitter graphs.** We filter Twitter data for specific hashtags and extract keywords from each tweet. For each hashtag, we use roughly 500 tweets to construct a bipartite graph with "players" representing advertisements and "items" representing keywords. The valuation of the keyword is determined

by the length of the tweet and the popularity of the keyword. We focus on four different hashtags, each used to construct a different graph: #ad (G5), #spon (G6), #giveaway (G7) and #win (G8).

**OXS valuations.** To analyze the spectrum of unit-demand and additivity, we construct the following OXS valuations. We fix the number of $n = 1024$ items in the ground set. An item of type $i$, $a_j$, has value $v(a_j) = i$. We vary the number of item types of items by parameter $m$, where there are $n/m$ items of each type and each $m$ yields a different OXS valuation. In our construction, all players have the same valuation. In one extreme, for $m = 1$, there is one item type and all 1024 items have value 1, which represents a unit-demand valuation. In the other extreme, $m = 1024$ so that there is exactly one item of each type, which represents an additive function.

## 5.2 Experimental Results

**General performance.** Overall, on our synthetic graphs, GSAS (blue) performed near optimally and outperformed the baselines of TRIMMED GREEDY, TOP $k$, LEARNED OPT and RANDOM (Figures 1a, 1b, 1c, 1d). It was able to obtain high value in much fewer rounds than the traditional GREEDY, which can be seen in the gap of performance between GSAS and TRIMMED GREEDY. We note that LEARNED OPT essentially learns a constant function and cannot distinguish between elements, which causes it to perform similarly to RANDOM.

On the Twitter data, we attempt to select certain advertising tweets that value keywords highly to maximize revenue for an ad placement agency. Sample keywords that were contained in selected ads are listed in Figure 2. We found that for smaller $k$, GSAS needed as many rounds as GREEDY to terminate so that the performance of both algorithms is near equivalent for $k$ smaller than 80. However, for larger values of $k$, GSAS terminated in much fewer rounds (Figures 1e, 1f, 1g, 1h).

On Twitter datasets, the oracle call to calculate the OXS valuation is computationally expensive as it includes a maximal weight matching step. Due to computation constraints, we do not use ADAPTIVE SAMPLING, which requires many oracle calls, as a benchmark. In Figure 2, we show the inferior performance of ADAPTIVE SAMPLING compared to GSAS on one such Twitter graph.

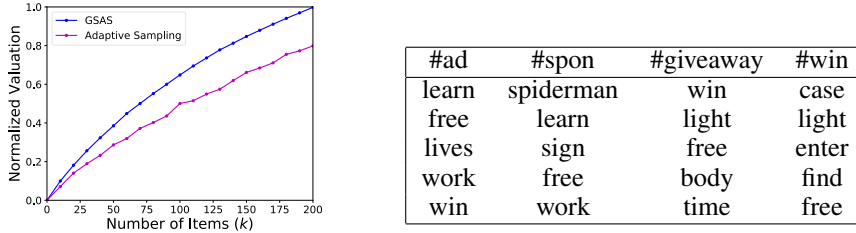

| #ad | #spon | #giveaway | #win |
|------|-----------|-----------|-------|
| learn | spiderman | win | case |
| free | learn | light | light |
| lives | sign | free | enter |
| work | free | body | find |
| win | work | time | free |

Figure 2: On the left, GSAS outperforms ADAPTIVE SAMPLING on a Twitter graph. Top keywords for each hashtag are listed on the right.

**Fewer number of rounds.** We note that in our experimental results, the true number of rounds needed for GSAS to terminate is much lower than the theoretical one of $\log(n)/\epsilon^3$. First, in order to reach value of $t = \epsilon OPT/k$, only $\log(\epsilon^{-1})/\epsilon$ rounds are needed. Second, in the inner loop, the factor of $\log(n)/\epsilon$ is an upper bound and adding more elements into the solution set results in fewer rounds. Additionally, we found that the outer iteration terminated prematurely when the solution set reached $k$ elements. These empirical observations show that GSAS can be much more computationally efficient than GREEDY. Even so, the number of rounds preformed by GSAS was quiet low and presented an improvement in the number of needed rounds. We elaborate on it in Appendix F.

**Spectrum of UD-additivity** In the two extremes where the OXS valuation is strictly additive or unit-demand (UD), TOP-K performs optimally by selecting the elements with the highest marginal contribution to the empty set in one round. In the general case where the objective valuation lies in the spectrum of UD-additivity, we found that GSAS outperforms its baselines in all regimes. In Figure 3, we normalize all values to the optimal solution as computed by GREEDY. Algorithms that perform better have values close to 1 in the figure.

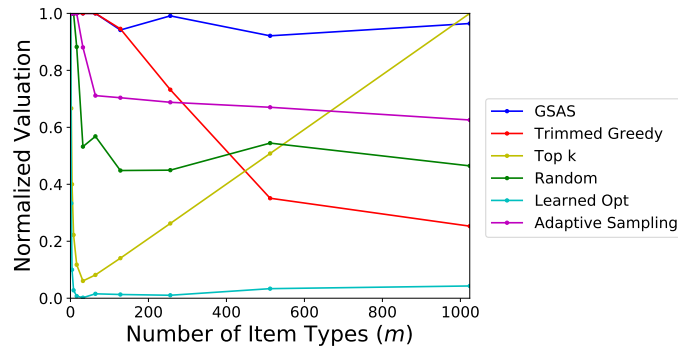
Figure 3: Performance of algorithms on UD-additive valuations.

## Broader Impact

This work focuses on the adaptivity of maximizing gross substitutes functions. While previous work has been done on maximizing submodular functions, a superclass of gross substitutes, little is known about the adaptivity complexity to achieve optimal results for this particular class of functions. Our results show an exponentially faster algorithm with near-optimal approximation guarantees for optimization of gross substitute valuations, which have numerous applications in microeconomics and market design [2, 33, 3, 23, 25] and appear in multiple fields such as discrete mathematics [28] and number theory [14].

The algorithm presented in this work is particularly relevant to applications on large datasets where sequential algorithms such as GREEDY become impractical and computationally infeasible. By using a low-adaptivity algorithm such as GSAS, we are able to take advantage of parallelization and dramatically speed up computation on large datasets. In Section 5, we show an application of this algorithm on large constructed Twitter networks to efficiently match keywords in advertisements to bidders or advertisers. In our experimental results, we show both the effective performance and computational efficiency of using GSAS on different networks. This shows that the limited adaptivity of GSAS can be effectively leveraged to analyze trends on other large-scale social networks and applications.

**Acknowledgments**
Ron Kupfer - This project has received funding from the European Research Council (ERC) under the European Union's Horizon 2020 research and innovation program (grant agreement No. 740282).
Yaron Singer - This research was supported by BSF grant 2014389, NSF grant CAREER CCF-1452961, NSF USICCS proposal 1540428, Google research award, and a Facebook research award.

## Footnotes

[1] In the context of Walrasian equilibrium, gross substitutes correspond exactly to the class of functions for which the greedy algorithm is optimal for all price vectors [31].

[2]For submodular functions this algorithm obtains a $1 - 1/e - \mathcal{O}(\epsilon)$ approximation to the optimal solution. We give the analysis in Appendix D.4

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
