[Supplementary Material]

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

# A  Constructions

## A.1  The Greedy Algorithm is Not Robust to Noise for Submodular Functions

For general submodular functions, it is not always true that the greedy algorithm is robust to noise. Consider the following example from [34]:

$$f(\emptyset) = 0$$
$$f(\{a\}) = 1, \ f(\{b\}) = 1, \ f(\{c\}) = 1 - \epsilon$$
$$f(\{a,b\}) = 2, \ f(\{a,c\}) = 1.5 - \tfrac{\epsilon}{2}, \ f(\{b,c\}) = 1.5 - \tfrac{\epsilon}{2}$$
$$f(\{a,b,c\}) = 2$$

For $k = 2$, the gap ratio between the noisy and clean versions is $\frac{3}{4}$. In general, it can be as small as $1 - \frac{1}{e}$.

## A.2  ADAPTIVE SAMPLING Fails to Maximize Gross Substitutes Functions

In [6], Balkanski et. al. presented an adaptive algorithm for submodular maximization. Similar to the greedy algorithm, this algorithm guarantees a $1 - \frac{1}{e}$ approximation to the optimal value. However, unlike the greedy algorithm, this algorithm does not guarantee an optimal solution for gross substitutes functions.

In this section, we show an example of an OXS function which is difficult for ADAPTIVE SAMPLING when $k$, the number of elements we wish to select, is small compared to the size of the ground set $n$.

**The construction.**  We consider the family of partitions $\mathcal{P}$ where the $n$ elements are partitioned into four disjoint sets: $A, B, C$ and $D$. Formally:

$$\mathcal{P} := \left\{ P = (A, B, C, D) : |A| = \frac{k}{3}, \ |B| = |C| = |D| = (n - \frac{k}{3})/3 \right\}.$$

We construct OXS functions $f_P$ which depend on a partition $P \in \mathcal{P}$, so the hard family of OXS functions is

$$\mathcal{F}_1 = \{f_P : P \in \mathcal{P}\}.$$

Given a partition $P = (A, B, C, D)$, we define $\frac{4k}{3}$ unit-demand functions. Those are $a_i, b_i, c_i$ and $d_i$ for $i \in [\frac{k}{3}]$ where

$$a_i(S) = 12 \cdot \mathbb{1}_{|S \cap A| > 0} \quad b_i(S) = 9 \cdot \mathbb{1}_{|S \cap B| > 0} \quad c_i(S) = 6 \cdot \mathbb{1}_{|S \cap C| > 0} \quad d_i(S) = 4 \cdot \mathbb{1}_{|S \cap D| > 0}.$$

The function $f_P$ is then defined to be the OXS function over these unit-demand functions.

Given cardinality constraint $k$, $\mathtt{OPT} = 9k$ simply by taking $k/3$ elements from $A$, $B$ and $C$.

For the ADAPTIVE SAMPLING algorithm, assume the number of rounds $r = k$ (which gives the algorithm the highest value). At each round, the algorithm looks for a random set $T$ of size $k/r = 1$ such that

$$f_S(T) = (\mathtt{OPT} - f(S))/k$$

where $S$ is the current solution set. In the first round, $S = \emptyset$ and randomly sampled sets should have a marginal contribution of at least $\frac{\mathtt{OPT}}{k} = 9$. Since there are very few elements from set $A$ compared to the elements in set $B$ which also have acceptable value, almost none of $A$ will be selected at this step. After $\frac{k}{3}$ elements from set $B$ are added to the solution set, the threshold is now $\frac{\mathtt{OPT} - f(S)}{k} = (9k - 9 \cdot \frac{k}{3})/k$. For the same reason as before, the algorithm will now add elements from set $C$ to the solution set. After adding another $\frac{k}{3}$ elements, the threshold is now $\frac{\mathtt{OPT} - f(S)}{k} = (9k - 9 \cdot \frac{k}{3} - 6 \cdot \frac{k}{3})/k = 4$. In the last round, $\frac{k}{3}$ elements from set $D$ are have good marginal contribution and with high probability the total value obtained at the end of ADAPTIVE SAMPLING is $\frac{19k}{3}$.

# B Missing Proofs for IMPATIENT GREEDY (Section 3.1)

## B.1 The First Stage of IMPATIENT GREEDY Selects Few Elements

**Lemma 3.** *Let $f : 2^N \to \mathbb{R}$ be a submodular function, $t = \frac{OPT}{\epsilon k}$ and $S$ be the set maintained at the end of the first while loop of* IMPATIENT GREEDY. *Then, $|S| \leq \epsilon k$.*

*Proof.* Denote by $(a_1, ... a_{|S|})$ the elements of $S$ in some order. Since $t = \frac{OPT}{\epsilon k}$, there are at most $\epsilon k$ elements such that $f_{\{a_1,...,a_{i-1}\}}(a_i) \geq t$ otherwise we have exceeded OPT by selecting this set. $\square$

## B.2 Proof of Theorem 1

*Proof.* Let set $S_1$ be the set of size $m \leq \epsilon k$ selected in the first while loop and $S_2 = S \setminus S_1$. Let $g(A) = f_{S_1}(A)$ and $O$ be the optimal solution for maximizing $f$ with cardinality constraint of $k - m$. Since $g$ is a gross substitutes function, $S_2$ is known to be optimal for maximization. Thus,

$$f(S) = f(S_1) + g(S_2) \geq f(S_1) + g(O) = f(S_1 \cup O).$$

By monotonicity, $f(S_1 \cup O) \geq f(O)$ and since $f(O)$ is a $\frac{k-m}{k}$ approximation to OPT we get:

$$f(S) \geq \frac{k-m}{k} OPT \geq \frac{k - \epsilon k}{k} OPT = (1 - \epsilon) OPT. \qquad \square$$

## B.3 IMPATIENT GREEDY for Submodular Functions

**Theorem 6.** *Given a monotone submodular $f : 2^N \to \mathbb{R}$,* IMPATIENT GREEDY *with threshold $t = \frac{OPT}{\epsilon k}$ returns a set $S$ such that $f(S) \geq (1 - 1/e - \epsilon) OPT$.*

*Proof.* As before, let set $S_1$ be the set of size $m \leq \epsilon k$ selected in the first while loop and $S_2 = S \setminus S_1$. Let $g(A) = f_{S_1}(A)$ and $O$ be the optimal solution of size $k - m$ for $f$. Since $g$ is a monotone submodular function, $S_2$ is known to guarantee a $1 - \frac{1}{e}$ approximation for its maximization problem. Thus,

$$f(S) = f(S_1 \cup S_2) = f(S_1) + g(S_2) \geq f(S_1) + (1 - e^{-1})g(O) = (1 - e^{-1})(f(S_1) + f_{S_1}(O)).$$

By the monotonicity assumption, $f_O(S_1) \geq 0$ and since $f(O)$ is a $\frac{k-m}{k}$ approximation to OPT we get that

$$f(S_1) + f_{S_1}(O) = f(O) + f_O(S_1) \geq f(O) \geq (1 - \epsilon) OPT.$$

Combining the above results, $f(S) \geq (1 - e^{-1} - \epsilon) OPT$. $\square$

# C Missing Proofs for STOCHASTIC GREEDY (Section 3.2)

## C.1 Proof of Lemmas for Theorem 2

We first prove the following lemma to compare the solution $S$ to the optimal solution.

**Lemma 4.** *Given a gross substitutes function $f : 2^N \to \mathbb{R}$ and two sets $S, T$ s.t. $|S| < |T|$, then $f(S) + f(T) \leq \max_{t \in T \setminus S} \{f(S \cup t) + f(T \setminus t)\}$.*

*Proof.* We define another function $f'$ on $n + |T| - |S|$ items by adding a set $D$ of size $|T| - |S|$ of dummy items: $f'(A \cup R) = f(A)$ for all $R \subseteq D$, where adding $R$ does not affect the function value. It is straightforward to verify that $f'$ is a gross substitutes function. Let $S' = S \cup D$ so $|S'| = |T|$. By Lemma 1 over $f'$, $S', T$ and $s \in D \subseteq S'$, we get

$$f(S) + f(T) = f'(S') + f'(T) \leq \max_{t \in T \setminus S'} \{f'(S' \cup t \setminus s) + f'(T \cup s \setminus t)\} = \max_{t \in T \setminus S} \{f(S \cup t) + f(T \setminus t)\}.$$

$\square$

**Lemma 5.** *Let $O_i = \{o_1, \ldots, o_i\}$ be the current solution of the greedy algorithm at iteration $i$ (recall that greedy is optimal for gross substitutes) and $S_i = \{a_1, \ldots, a_i\}$ be any set of size $i$. Then, for all $i < k$, there exists $o \in O_{i+1}$ s.t. $f_{O_i}(o_{i+1}) \leq f_{S_i}(o)$.*

*Proof.* By Lemma 4 there exists $o \in O_{i+1}$ such that $f(S_i) + f(O_{i+1}) \leq f(S_i \cup o) + f(O_{i+1} \setminus o)$. Rearranging yields the result:

$$f_{O_i}(o_{i+1}) = f(O_{i+1}) - f(O_i) \leq f(O_{i+1}) - f(O_{i+1} \setminus o) \leq f(S_i \cup o) - f(S_i) = f_{S_i}(o)$$

where the first inequality is due to the optimality of $O_i$ and the second from the above inequality. □

## C.2 Proof of Theorem 2

*Proof.* Observe that

$$\mathbb{E}[f(S)] = \mathbb{E}[\sum_{i \leq k} f_{S_{i-1}}(s_i)] \geq \mathbb{E}[\sum_{i \leq k} \xi_i f_{O_{i-1}}(o_i) - \zeta_i] = \sum_{i \leq k} (\mathbb{E}[\xi_i] f_{O_{i-1}}(o_i) - \mathbb{E}[\zeta_i])$$

$$\geq \sum_{i \leq k} \hat{\xi} f_{O_{i-1}}(o_i) - \hat{\zeta} = \hat{\xi} \texttt{OPT} - \hat{\zeta}$$

where the first inequality follows from Lemma 5 and the second from the definitions of $\hat{\xi}$ and $\hat{\zeta}$. □

# D Missing Proofs for GSAS (Section 3.3)

## D.1 Proof for Lemma 2

*Proof.* The outer loop runs $\Delta$ times. Each inner iteration makes at most $\log(n)/\epsilon$ rounds since at each round, $|X|$ decreases by at least a factor of $1 - \epsilon$. Finding $i^*$ can be done in a single round. □

## D.2 Lemmas for the Proof of Theorem 3

**Lemma 6.** *Given a submodular function $f : 2^N \to \mathbb{R}$, let $S$ be the set at the end of the first while loop of* GSAS. *Then, either $f(S) = \texttt{OPT}$ or $|S| < 3\epsilon k$ with probability $1 - o(e^{\epsilon k/8})$.*

*Proof.* Denote by $(a_1, ... a_{|S|})$ the elements of $S$ in some order. Since $t^\star$ is a downward discretization of $\frac{\texttt{OPT}}{\epsilon k}$, we have that $t^\star \geq \frac{\texttt{OPT}}{(1+\epsilon)\epsilon k}$ and there are at most $(1 + \epsilon)\epsilon k$ elements such that $f_{\{a_1,...,a_{i-1}\}}(a_i) \geq t^\star$ (otherwise we have exceeded $\texttt{OPT}$ by selecting this set). However, we still need to show that the randomization of the algorithm does not choose too many items with marginals contributions below threshold $t$. Let $a_1, ..., a_l$ be the elements chosen in the process. In order to estimate the number of selected elements such that $f_{\{a_1,...,a_{i-1}\}}(a_i) < t^\star$, we first note that at most $k$ elements are chosen in total. By definition of $i^*$, for all $i < i^*$, each item $a_i$ has a probability of at least $1 - \epsilon$ to maintain $f_{\{a_1,...,a_{i-1}\}}(a_i) \geq t^\star$. Since each element in the sequence is drawn independently of previous selections, using the Chernoff bound, with probability at least $1 - e^{-\frac{\epsilon k}{8}}$, no more than $1.5\epsilon k$ such elements violate this condition. Thus, at the end of the first iteration, at most $((1 + \epsilon) + 1.5)\epsilon$ elements were chosen and $|S| < 3\epsilon k$. □

In the following executions of the outer loop, there are no elements with marginal contribution exceeding $\frac{\texttt{OPT}}{\epsilon k}$ and the algorithm can be reduced to the original version in [7] on $f$ where $S$ is chosen in the first iteration. Below, we show analogous properties but with slight adjustments.

**Lemma 7.** *Let $f : 2^N \to \mathbb{R}$ be a submodular function. At the end of any inner iteration, $t \geq (1 - \epsilon) \max_a f_S(a)$.*

*Proof.* At the end of the first iteration, $X = \emptyset$. So, for all $a \in N$, $a$ was discarded from $X$ at some previous inner iteration with current solution $S'$ such that $f_{S' \cup \{a_1,...,a_{i^*}\}}(a) < t$ and $S' \cup \{a_1, ..., a_{i^*}\} \subseteq S$. Thus, by submodularity, $f_S(a) < t$. That is, $t \geq (1 - \epsilon) \max_a f_S(a)$ and the property holds at this point.

We now show that this property is maintained through the algorithm when either $S$ or $t$ is updated: First, since we only add elements to $S$, the property is unaffected by submodularity. Second, assume we update $t' = (1 - \epsilon)t$. By the same claim as in the first iteration, $t \geq (1 - \epsilon) \max_a f_S(a)$. Since $t' = (1 - \epsilon)t$, we get the desired result. □

Lastly, we show that after the first iteration, each element added to the solution by the algorithm is a stochastic greedy step.

**Lemma 8.** *Let* $a_1, ..., a_{k^*}$ *be a random sequence during any iteration of the outer-loop except the first. Then, for all* $i < i^*$,

$$\mathbb{E}_{a_i}[f_{S \cup \{a_1,...,a_{i-1}\}}(a_i)] \geq (1 - \epsilon)^2 \max_a f_{S \cup \{a_1,...,a_{i-1}\}}(a)$$

*Proof.* Observe that

$$\mathbb{E}_{a_i}[f_{S \cup \{a_1,...,a_{i-1}\}}(a_i)] \geq \Pr\left(f_{S \cup \{a_1,...,a_{i-1}\}}(a_i) \geq t\right) \cdot t = \frac{|X_{i-1}|}{|X|} \cdot t \geq (1-\epsilon)^2 \max_a f_{S \cup \{a_1,...,a_{i-1}\}}(a)$$

where the first inequality is by definition of expectation, and the second since $i \leq i^\star$ and by Lemma 7. □

### D.3 Proof of Theorem 3

*Proof.* By Lemma 8, GSAS behaves similarly to the STOCHASTIC GREEDY with noise distribution $\mathcal{D}_i$ where $\zeta_i = 0$ and $\mathbb{E}[\xi_i] \geq (1 - \epsilon)^2$. The key differences of GSAS are (1) the initial threshold may be too low and (2) the termination of the algorithm before $k$ elements are selected.

We start by handling the first issue. Let $S_1$ be the set of size $m$ selected by GSAS in the first round, and $g(A) = f(S_1 \cup A)$ be the conditioned function, $O$ be an optimal solution of the maximization of $g$ under cardinality constraint $k - m$ and $G$ be the solution returned by STOCHASTIC GREEDY for $g$ (under cardinality constraint $k - m$) with $\mathbb{E}[\xi_i] \geq 1 - 2\epsilon$ and $\zeta_i = 0$ for all $i$.

By Lemma 8 we have that $\mathbb{E}[g(S)] \geq (1 - 2\epsilon)g(G)$ and by Theorem 2, we have $\mathbb{E}[g(G)] \geq (1 - \epsilon)g(O)$. Together, we get that

$$\mathbb{E}[g(S)] \geq (1 - 3\epsilon)g(O).$$

Since the greedy algorithm is optimal for gross substitutes functions and $g$ is gross substitutes, we have that $g(O)$ is no less than value of any set of that size. In particular, this is true for the set returned by the greedy algorithm on $f$ with $k - m$ cardinality constraint. That is, $g(O) \geq \text{OPT}_{k-m}$ where $\text{OPT}_{k-m}$ is the optimal value of $f$ with cardinality constraint $k - m$. Combining the above properties yield

$$\mathbb{E}[g(S)] \geq (1 - 3\epsilon)g(O) \geq (1 - 3\epsilon)\text{OPT}_{k-m} \geq (1 - 3\epsilon)\frac{k - m}{k}\text{OPT}.$$

By Lemma 6, we have that $m = 3\epsilon k$ and we get that $S$ is a $1 - 6\epsilon$ approximation.

Next we handle the possibility that GSAS terminates with fewer than $k$ items. Since GSAS terminates after $\Delta$ iterations with $t < \frac{\epsilon\text{OPT}}{k}$, we may not obtain $k$ elements. However, we miss at most $k$ elements where each element contributes at most $t$ to the solution set. Hence, there is a loss of at most $tk = \epsilon\text{OPT}$ of the total value. Thus, we get that GSAS gives a $1 - 7\epsilon$ approximation. □

### D.4 GSAS Obtains a $1 - 1/e - \mathcal{O}(\epsilon)$ Approximation for Submodular Functions

**Theorem 7.** *Given a monotone submodular function* $f : 2^N \to \mathbb{R}$ *and for any* $\epsilon > 0$, GSAS *returns a set* $S$ *such that* $\mathbb{E}[f(S)] \geq (1 - 1/e - \mathcal{O}(\epsilon))\text{OPT}$ *approximation ratio using* $\mathcal{O}(\log(n)/\epsilon^3)$ *rounds.*

*Proof.* Let set $S_1$ be the set of size $m$ selected by GSAS in the first round. Let $g(A) = f(S_1 \cup A)$ be a submodular function, $A$ be the solution of size $k - m$ for $g$ returned by the greedy algorithm and $G$ be the solution of size $k - m$ for $g$ returned by STOCHASTIC GREEDY with $\mathbb{E}[\xi_i] \geq 1 - 2\epsilon$ and $\zeta_i = 0$ for all $i$. Let $S$ be the result of GSAS.

By Theorem 8 we know that $g(S) \geq (1 - 2\epsilon)g(G)$. By Theorem 2, we have $\mathbb{E}[g(G)] \geq (1 - \epsilon)g(A)$. Combining these together, we get $\mathbb{E}[g(S)] \geq (1 - \mathcal{O}(\epsilon))g(A)$.

By [22], we know that the greedy algorithm returns a $1 - e^{-(1-\epsilon)}$ approximation to the optimal value of $g$. By submodularity, we know that the returned value is at least as good as the greedy algorithm

on $f$ with $k - m$ cardinality constraint by monotonicity of $f$. That is $g(A) \geq (1 - e^{-(1-\epsilon)})\text{OPT}_{k-m}$ where $\text{OPT}_{k-m}$ is the optimal value of $f$ with cardinality constraint $k - m$. Together,

$$\mathbb{E}[g(S)] \geq (1 - 3\epsilon)g(A) \geq (1 - 3\epsilon)(1 - e^{-(1-\epsilon)})\text{OPT}_{k-m} \geq (1 - 3\epsilon)(1 - e^{-(1-\epsilon)})\frac{k-m}{k}\text{OPT}$$

Setting $m = 3\epsilon k$ as Lemma 6 suggests, we get a $1 - 1/e - \mathcal{O}(\epsilon)$ approximation.

Finally, since GSAS terminates after $\Delta$ iterations with $t = \frac{\text{OPT}}{\epsilon k}$, we may not reach a set of size $k$. However, we miss at most $k$ elements which contribute value at most $tk$. Thus, we get a $1 - 1/e - \mathcal{O}(\epsilon)$ approximation. □

## E   Missing Proofs from Section 4

### E.1   Warm-up: Hardness of Optimization in One Round

Before showing the two main lower bounds, we show that there is no 1-adaptive algorithm that obtains a constant approximation for maximizing OXS functions when the queries are of size $\mathcal{O}(k)$. At a high level, we construct a family of functions $\mathcal{F}$ where each function $f_P \in \mathcal{F}$ is defined by a partition $P$ of the ground set of elements $N$. The central argument for the analysis is that an algorithm cannot learn the partition $P$ from one round of poly-many queries $f_P(S)$ and that the algorithm needs to know $P$ to find a solution $S$ with high value $f_P(S)$.

**The construction.**   We consider the family of partitions $\mathcal{P}$ where the $n$ elements are partitioned into three disjoint sets: the set $G$ of good elements of size $k = |G| = n^{\frac{1}{3}}$, the set $B$ of bad elements of size $n^{\frac{2}{3}}$, and the set $M$ of masking elements of size $n - |G| - |B|$. Formally:

$$\mathcal{P} := \left\{ P = (G, B, M) : |G| = n^{\frac{1}{3}}, |B| = n^{\frac{2}{3}}, |M| = n - |G| - |B| \right\}.$$

We construct OXS functions $f_P$ which depend on a partition $P \in \mathcal{P}$, so the hard family of OXS functions is $\mathcal{F}_1 = \{f_P : P \in \mathcal{P}\}$. Given a partition $P = (G, B, M)$, we define $n^{\frac{1}{3}} + \frac{n^\epsilon}{2}$ unit-demand functions. Those are $g_i$ for $i \in [n^{\frac{1}{3}}]$ and $b_i$ for $i \in [\frac{n^\epsilon}{2}]$ where

$$g_i(S) = \mathbb{1}_{|S \cap G| > 0} \quad \text{and} \quad b_i(S) = \mathbb{1}_{|S \cap B| > 0}.$$

The function $f_P$ is then defined to be the OXS function over unit-demand functions $g_i, b_i$. Note that $f_P(G) = n^{1/3}$, $f_P(B) = n^\epsilon/2$, and $f_P(M) = 0$. We obtain the following lower bound for the family of OXS functions $\mathcal{F}_1$.

**Theorem 8.** *There is no 1-adaptive algorithm that obtains, with probability $\omega(\frac{1}{n})$, an $n^{-\frac{1}{3}+\epsilon}$ approximation for maximizing OXS functions under a cardinality constraint when the queries are sets of size $\mathcal{O}(k)$, for any constant $\epsilon > 0$.*

*Proof.* First, we show that for any constant $c > 0$, good and bad elements are separable with exponentially small probability when using sample sizes smaller than $ck$. Since we take a sample of size $t \leq ck$, the probability for seeing more than $d$ elements from either $G$ or $B$ is small:

$$\Pr(|(G \cup B) \cap S| > d) < \binom{t}{d} \left( \frac{|G \cup B|}{n} \right)^d$$

$$< \binom{ck}{d} \left( \frac{2n^{\frac{2}{3}}}{n} \right)^d \leq \left( \frac{2ecn^{\frac{1}{3}}n^{\frac{2}{3}}}{nd} \right)^d = \left( \frac{2ec}{d} \right)^d$$

where the last inequality follows from $\binom{a}{b} \leq \left( \frac{ae}{b} \right)^b$ where $e$ is Euler's number. By taking $d = \frac{n^\epsilon}{2}$, we get an exponentially small probability of $|(G \cup B) \cap S| > d$. Thus, the marginal value of good or bad elements equals 1 based on the constructed OXS function.

At the end of the sampling stage, we can assume that $g_i$ and $b_i$ are indistinguishable with high probability. Under this assumption, we can bound the probability for selecting more than $d$ objects from $G$ Since the algorithm chooses up to $k$ elements from $G \cup B$ at random:

$$P(|G \cap S| > d) < \binom{k}{d} \left( \frac{|G|}{|G \cup B|} \right)^d < \binom{n^{\frac{1}{3}}}{d} \left( \frac{n^{\frac{1}{3}}}{n^{\frac{2}{3}}} \right)^d < d^{-d}$$

By taking $d = \frac{n^\epsilon}{2}$ for large enough $n$, we get an exponentially small probability for selecting more than $d$ elements from $G$. So any algorithm selects w.h.p. at most $\frac{n^\epsilon}{2}$ elements from $G$ and the rest are from $B$ and $M$ which contribute at most $\frac{n^\epsilon}{2}$. Thus, we get that w.h.p. $f(S) < n^\epsilon$. Since $\texttt{OPT} = n^{\frac{1}{3}}$ the claim holds. $\qquad\square$

### E.2 Analysis for Hardness of Non-Adaptive Algorithms

**The construction.** We consider the family of partitions $\mathcal{P}$ that consists of all partitions $P$ where the $n$ elements are partitioned into three disjoint sets as follows: set $G$ of good elements of size $\log^3 n$, set $B$ of bad elements of size $\log^6 n$, and set $M$ of masking elements of size $n - |G| - |B|$.

We construct OXS functions $f_P$ which depend on a partition $P \in \mathcal{P}$, so the hard family of OXS functions is $\mathcal{F} = \{f_P : P \in \mathcal{P}\}$. Given a partition $P$, we define unit-demand functions function $u_i$ for $i \in [|G| + |B|]$, $v_i$ for $i \in [\log n]$, and $w_i$ for $i \in [\log^3 n]$, where

$$u_i(S) = \mathbb{1}_{|S \cap (G \cup B)| > 0}, \qquad v_i(S) = 2_{|S \cap (G \cup B)| > 0}, \qquad w_i(S) = \max\left(2_{|S \cap G| > 0}, \mathbb{1}_{|S \cap M| > 0}\right).$$

The function $f_P$ is then defined to be the OXS functions over unit-demand functions $u_i, v_i, w_i$.

**The analysis.** We consider a uniformly random function $f_P \in \mathcal{F}$. The main lemma is that elements in $G$ and $B$ are indistinguishable from one round of queries over any (potentially non-feasible) sets. The main observation to obtain the indistinguishability of $G$ and $B$ over large sets is that if set $S$ contains sufficiently many masking elements $M$, then the marginal contribution of a good element to $S$ is 1 due to $w_i$, which is equal to the marginal contribution of a bad element when $S$ contains sufficiently many elements form $G \cup B$. If a set $S$ contains no masking elements and at least $\log n$ elements from $G \cup B$, then good and bad elements have marginal contributions that are 2 and 1 respectively, which leads to the following.

**Lemma 9.** *Consider any algorithm $\mathcal{A}$ and a uniformly random function $f_P \in \mathcal{F}_r$. For any set $S$, $f_P(S)$ is independent of the partition of $G \cup B$ into $G$ and $B$ with probability $1 - n^{-\omega(1)}$.*

*Proof.* We split the analysis into two parts.

**For small sample sizes:** If we take a sample of size $t \leq \log^3(n)$ the probability of seeing more than $\log(n)$ elements from either $G$ or $B$ is super-polynomially small:

$$\Pr(|(G \cup B) \cap S| > \log(n)) < \binom{t}{\log(n)} \left(\frac{|G \cup B|}{n}\right)^{\log(n)}$$

$$< \binom{\log^3(n)}{\log(n)} \left(\frac{2\log^6(n)}{n}\right)^{\log(n)} < \left(\frac{2\log^9(n)}{n}\right)^{\log(n)} < n^{-\log(\log(n))}$$

So w.h.p., the marginal contribution of good or bad elements is 2.

**For large sample sizes:** If we take a sample of size $t \geq \log^3(n)$, then the probability of seeing less than $\log^2(n)$ masking elements from set $M$ is super-polynomially small:

$$\Pr(|M \cap S| < \log^2(n)) < \binom{t}{t - \log^2(n)} \left(\frac{|G \cup B|}{n}\right)^{t - \log^2(n)}$$

$$< \binom{n}{\log^2(n)} \left(\frac{\log^7(n)}{n}\right)^{\frac{\log^3(n)}{2}} < n^{\log^2(n)} \left(\frac{\log^7(n)}{n}\right)^{\frac{\log^3(n)}{2}}$$

$$= n^{-\log^3(n)/2 + \log^2(n) + 7\log^2(n)\log(\log(n))/2} < n^{-\log(n)}.$$

where the first inequality follows from bounding the probability that at least $t - \log^2(n)$ elements are in $G \cup B$ and the last inequality holds for large $n$. When the number of masking elements exceeds $\log^2(n)$, $g_i$ and $b_i$ are indistinguishable. $\qquad\square$

### E.2.1 Proof of Theorem 4

*Proof.* From Lemma 9, we may assume that w.h.p. $g_i$ and $b_i$ are indistinguishable. Since the algorithm may choose at most $k$ elements from $G \cup B$ at random, we can bound the probability for selecting more than $\log(n)$ elements from $G$:

$$P(|G \cap S| > \log(n)) < \binom{k}{\log(n)} \left( \frac{|G|}{|G \cup B|} \right)^{\log(n)}$$

$$< \binom{\log^2(n)}{\log(n)} \left( \frac{\log^3(n)}{\log^6(n) + \log^3(n)} \right)^{\log(n)} < \left( \frac{\log^5(n)}{\log^6(n)} \right)^{\log(n)} = \log(n)^{-\log(n)} = n^{-\log(\log(n))}$$

So, for any $\delta > 0$ and large enough $n$, we get that with high probability $f(S) \le \log^2(n) + 2\log(n)$ and $(2 - \delta)f(S) < \texttt{OPT}$. □

### E.3 Analysis for Hardness of $\tilde{o}(\log n)$ Rounds of Adaptivity

**The construction.** We consider the family of partitions $\mathcal{P}$ that consists of all partitions $P$ such that the $n$ elements are partitioned into $r + 2$ disjoint sets, where $r = \frac{\log n}{4 \log(\log n)} - 1$, as follows: the set $G$ of good elements of size $k = \log^4 n$, sets $B_i$ of bad elements of size $\log^{4(i+1)} n$ for $i \in [r]$, and the remaining elements are dummy elements comprising set $M$.

We construct OXS functions $f_P$ which depend on a partition $P \in \mathcal{P}$, so the hard family of OXS functions is $\mathcal{F}_r = \{f_P : P \in \mathcal{P}\}$. Given a partition $P$, we define $\log^4 n + r \log^2 n$ unit-demand functions function $g_i$, for $i \in [\log^4 n]$, and $b_{i,j}$, for $i \in [r]$ and $j \in [\log^2 n]$, where

$$g_i(S) = \mathbb{1}_{|S \cap G| > 0} \quad \text{and} \quad b_j(S) = \mathbb{1}_{|S \cap B_i| > 0}.$$

The function $f_P$ is then defined to be the OXS function over unit-demand functions $g_i, b_{i,j}$. Similar to the previous construction, we note that the set $G$ has high value $f_P(G) = \log^4 n$ while the sets $B_i$ have low value $f_P(B_i) = \log^2 n$ for all $i \in [r]$. We also note that for all $i$, $|B_i| = \log^4 n |B_{i-1}|$.

**The analysis.** We consider a uniformly random function $f_P \in \mathcal{F}_r$. The main lemma for this construction is, informally, that the algorithm can learn at most one part $B_i$ of the partition $P$ per round. More precisely, let $B^i = \bigcup_{i' \le r-i} B_{i'}$. If the queries of the algorithms at round $i$ are chosen independently of the partition of $B^i \cup G$ into $B_1, \ldots, B_{r-i}, G$, then the algorithm may learn $B_{r-i}$, but cannot learn any information about the partition of $B^{i+1} \cup G$ into $B_1, \ldots, B_{r-(i+1)}, G$. Additionally, if the algorithm cannot distinguish elements in $B_1$ and $G$, then, with high probability, the algorithm returns a solution that is at best a $1/\log n$ approximation to $f_P(G)$.

**Lemma 10.** *Consider a uniformly random function $f_P \in \mathcal{F}_r$. Let $B^i = \bigcup_{i' \le r-i} B_{i'}$. For any collection of $\mathrm{poly}(n)$ queries $\{S_j\}_j$, each of size $\mathcal{O}(k)$, that are chosen independently of the partition of $B^i \cup G$ into $B_1, \ldots, B_{r-i}, G$, then, with probability smaller than $2^{-\log^2 n}$, for all $S_j$, the value $f_P(S_j)$ is independent of the partition of $B^{i+1} \cup G$ into $B_1, \ldots, B_{r-(i+1)}, G$.*

*Proof.* We show the claim by induction. Before the first round, we have no information. Assume we cannot distinguish between any element in $B^{i-1}$ and $G$. Any algorithm samples at most $ck$ elements from $B^i \cup G$ at random. The number of elements chosen from $B^i \cup G$ is bounded w.h.p.:

$$\Pr\left( |(G \cup B^i) \cap S| > \log^2(n) \right) < \binom{t}{\log^2(n)} \left( \frac{|G \cup B^i|}{|G \cup B^{i-1}|} \right)^{\log^2(n)}$$

$$< \binom{ck}{\log^2(n)} \left( \frac{1}{\log^3(n)} \right)^{\log^2(n)} \le \left( \frac{c \log^4(n)}{\log^5(n)} \right)^{\log^2(n)} < 2^{-\log^2(n)}$$

Thus, w.h.p. the marginal contribution of $B^i$ and $G$ is equal to 1 so the sets are indistinguishable. □

**Lemma 11.** *Consider any algorithm $\mathcal{A}$ and a uniformly random function $f_P \in \mathcal{F}_r$. If for all queries $S$ by $\mathcal{A}$, $f_P(S)$ is independent of the partition of $B_1 \cup G$ into $B_1$ and $G$, the probability of returning a set $S$ such that $f_P(S) > \frac{f_P(G)}{\log n}$ is smaller than $2^{-\log^2 n}$.*

*Proof.* Since the algorithm may choose up to $k$ elements from $G \cup B$ at random and with probability at least $1 - 2^{-\log^2(n)}$, $G$ and $B$ are indistinguishable, we can bound the probability for selecting more than $d$ elements from $G$:

$$P(|G \cap S| > d) < \binom{k}{d}\left(\frac{|G|}{|G \cup B|}\right)^d < \binom{\log^4(n)}{d}\left(\frac{1}{\log^3(n)}\right)^d < \left(\frac{\log(n)}{d}\right)^{-d}$$

Setting $d = \log^2(n)$, we get that at most $\log^2(n)$ elements from $G$ were selected. The contribution of all the other elements is bounded by $r \log^2(n)$ and thus, $f(S) < \log^3(n)$. Since $\texttt{OPT} = \log^4(n)$, the claim holds. $\square$

The proof of Theorem 5 follows from combining both Lemma 10 and Lemma 11.

# F   Experimental Constructions

In this section, we discuss in detail the construction of synthetic graphs G3 and G4 and Twitter graphs.

**Synthetic bipartite graphs.**   We generate randomized bipartite graphs with $m = 200$ players and $n = 200$ items, where the probability that an edge exists between a certain player and node is 0.25 for G3 and 0.75 for G4. This probability affects the number of neighbors the graph contains and the overall graph density. We select 75% of players and items to be "bad" nodes and the remaining node to be "good". We then insert an additional 50 item nodes into the graph as "masking" nodes and construct edges between these 50 nodes and 10 randomly chosen players and assign an edge weight of 1 to these edges. For the remaining edges, we assign low weights (generated uniformly from 0 and 0.1) to edges that contain any "bad" node and high weights (generated uniformly from 0.9 and 1) to edges that connect two "good" nodes. The edge weights were generated so that 60% of nodes had low weights and 40% of nodes had high weights.

**Twitter graphs.**   For a particular hashtag, we first filter for 500 tweets that contain that particular hashtag. We then use the NLP library RAKE to extract keywords from each tweet. We discard all tweets with one or less keyword, to ensure the graph is of a certain density. For each keyword, we then calculate the frequency of that word in the English corpus on the Zipf scale, where the frequency is the base-10 logarithm of the number of times it appears per billion words. Scores generally fall between 0 and 8. To assign weights to each edge, we multiply the number of keywords of a tweet by the Zipf score of the keyword.

**Number of rounds.**   For G1-G4 ($k = 10 : 100$ with leaps of 10) the number of rounds were: G1 Rounds = [18, 16, 14, 19, 17, 15, 20, 17, 15, 20] G2 Rounds = [3,3,4,5,6,7,8,9,16,16] G3 Rounds = [4, 6, 7, 8, 9, 10, 11, 13, 14, 16] G4 Rounds = [4, 6, 7, 8, 9, 10, 11, 13, 14, 16]. For $G5 - 8$ the number of rounds is approximately $k$ with $\epsilon = 0.1$ as seen in the plots by the quality of TRIMMED GREEDY. Using $\epsilon = 0.3$ reduces the number of rounds dramatically without harming the results, and the plots are essentially identical up to **Trimmed Greedy** which now have the lowest performances. For example, on data set G7 with $k = 150$ and $\epsilon = 0.1$, GSAS uses 138 rounds while 42 rounds are sufficient for $\epsilon = 0.3$ with the same returned value.