[Reviews · NeurIPS 2020]

Review 1

Summary and Contributions: The authors study the adaptive complexity of maximizing a gross substitutes set function subject to a cardinality constraints of at most k. Adaptive complexity refers to number of sequential rounds of computation, where in each round polynomial number of set function computations are made. They provide an algorithm with adaptive complexity of O(log n) that obtains a (1 - \eps) approximation. They also show that there is no 1-adaptive algorithm that obtains better than 1/2 + \eps approximation. There is also no \tilde{o}(log n) algorithm that can obtain (1 - \eps) approximation if restricted to queries for sets of size O(k). The authors also include empirical results that compare approximation obtained by their algorithm with a broad range of reasonable benchmarks. The empirical results also provide a real world example of a gross-substitute function that is not additive or unit-demand.

Strengths: Soundness of the claims: The authors fully justify their claim. While most of the proofs are in the appendix, authors provide a high level sketch with intuition in the main body of the paper. Significance and novelty of the contribution: Authors provide best adaptive algorithms for maximizing a gross-substitutes function subject to cardinality constraint. Gross-substitutes is an important class of set functions. The authors' results show that they have obtained the best bound possible. They also show a separation between Gross-substitutes and their super-class of Submodular valuation functions and Gross-substitutes and more commonly occurring subclasses of additive and unit-demand functions, thus firmly establishing the need for new techniques. The authors approach builds on previous techniques from two papers: 1) Balkanski, Rubinstein, Singer, STOC'19 - adaptive complexity for submodular under matroid constraint. 2) Hassidim, Singer, JMLR'2017 - proposes stochastic greedy for optimizing monotone submodular functions But neither technique by itself is sufficient. Their algorithm builds on the algorithm of 1) called "adaptive sequencing" but that by itself doesn't have optimal adaptive complexity. The authors pair it with the idea of an impatient greedy that uses the fact that greedy can provide the optimal for gross-substitutes. Another closely related work is: Rubinstein, Singer, STOC'18 - Adaptive Complexity for Submodular Optimization which provides a 1/3 approximation with an adaptive complexity of O(log n) for submodular valuations. The authors empirically compare with the algorithm of 2 and show a large gap. Relevance to the NeurIPS community: I think this work is highly relevant to the NeurIPS community. Optimization problems of the form max f(S) subject to |S| <= k often come up in the AI and theory literature. This work concerns the class of gross-substitute function which is an important generalization of additive and unit-demand and include OXS which as the authors empirical study shows can arise naturally in practical problems. The idea of adaptive complexity is motivated from the practical use-case where using parallelization we can carry out many more computations in fewer rounds.

Weaknesses: None.

Correctness: I did not verify all the proofs in the appendix, but the ones I did were satisfactory.

Clarity: The paper is very well written. Post-rebuttal: I do agree with the other reviewer that the description of the main algorithm is not fully precise. As the authors note in their rebuttal the reviewer's comments can be addressed. I would encourage the authors to make the description more precise and be sure to handle edge cases.

Relation to Prior Work: The authors attribute techniques to previous work when relevant. Explain why more previous techniques did not work - I found answers to all my critical thoughts.

Reproducibility: Yes

Additional Feedback:


Review 2

Summary and Contributions: This paper studies the adaptive complexity of maximizing a gross substitute function subject to a cardinality constraint. The task is to devise an algorithm that works in rounds, in each round, the algorithm can issue a polynomial number of value queries, and the goal is to find a set of at most k elements that approximately maximizes the function subject to the size constraints. The main result is an algorithm that runs in O(log n/eps^3) rounds and gets a 1-epsilon approximation. The following lower bounds are given: (1) There is no single-round algorithm that gives a 1/2+eps approximation. (2) Assuming that the algorithm queries sets of size O(k), there is no o(log n) algorithm that obtains a constant approx. The theoretical improvements of the proposed algorithm over prior work is validated with experiments on synthetic bipartite graphs and Twitter data.

Strengths: The greedy algorithm is optimal for GS functions, but has adaptive complexity n. Prior work has given a O(1) approximate O(log n) adaptive algorithm for monotone submodular fcts subject to matroid constraints. The results in this paper concern an important subclass of submodular - gross substitutes. And given the prior work the goal can only be a 1-eps approximate algorithm with o(n) rounds. The main result shows its indeed possible to get 1-eps with O(log n/eps^3) rounds. The analysis goes via nice and intuitive variants of the greedy algorithm. The performance improvements of the algorithm are validated on synthetic and real data.

Weaknesses: Something that goes a bit unnoticed in the description of the results is the switch from submodular + matroid to GS + cardinality (uniform matroid) constraints. Indeed, the techniques in this paper for showing the positive result build on [7] and are somewhat specific to cardinality constraints. The lower bound proof uses the classic idea of hiding a special set which a poly number of queries cannot identify, although one has to be a bit more careful. This is the reason the paper only gets a lower bound under the assumption that the algorithm only queries small sets. It would have been nice to see some experiments on even richer classes of gross substitute functions.

Correctness: As far as I could check the theoretical results seems sound, and the experimental setup seems to make sense.

Clarity: Yes, the paper is generally very well written, and clear as to what its conceptual, technical, and experimental contributions are.

Relation to Prior Work: By now there is quite some work on the adaptive complexity of monotone submodular functions subject to a matroid constraint. This paper addresses a special case (restricting submodular to gross substitutes and general matroids to uniform matroids), and shows stronger positive results, as well as novel impossibility results. The techniques are somewhat similar to prior work (in particular the STOC'19 paper by Balkanski et al.), and general techniques for showing lower bounds in query models. But also require some new ideas.

Reproducibility: Yes

Additional Feedback: N/A


Review 3

Summary and Contributions: Given a function f:2^[n]->R that satisfies the gross-substitutes condition and a cardinality constraint k, define OPT = max_{S subseteq [n], |S| \leq k} f(S). The main result of this paper is an algorithm that given epsilon finds a subset S s.t. f(S) \geq (1-O(epsilon))OPT and has an “adaptive complexity” of O(log n / epsilon^3). Adaptive complexity is a new notion suggested by a STOC 2018 paper and already quite extensively studied since then for optimizing a submodular function which is a superclass of gross-substitutes. An adaptive complexity of r means that the algorithm has r steps, each step makes a polynomial number of calls to f() which are “parallel” in the sense that the parameter given to f() may not depend on the result of other calls to f() in the same step. While I can imagine the motivation for this notion it would have been nice to see some discussion in the intro, maybe repeating the original motivation from STOC 2018 (after all this is really a new notion from a couple of years ago). Comments: 1. It could help the readers if you fix some misunderstandings that I had while reading algorithm 3: what happens if, in line 8, the set {i s.t. |X_i| < (1-epsilon)|X|} is empty? And shouldn’t X in line 4 receive N\S and not N? I also don’t manage to run very simple examples in my head and understand the execution of the algorithm on these examples. Suppose f() is additive, k items have value 1, N-k have value 1000, N is much larger than k, and epsilon=0.1. The first time we get to line 6, k^* = k and w.h.p. all the k values that we draw from X are 1’s. In the first round the threshold t^* will be very high (10,000) hence X_1 will be the empty set. As a result line 9 will add a low value item to S. I think this will continue the same way until S is filled with k low value items. I must be getting something wrong... My point is that the paper does not really explain the algorithm. The intuition to the proof by discussing the two simpler algorithms IMPATIENT GREEDY and STOCHASTIC GREEDY.is nice but I think there should also be a more elaborate discussion on the specifics of algorithm 3 itself. 2. One way to save space for this purpose (in my opinion) is to merge Section 4 with the more informal stuff in the intro about the lower bounds. I got the feeling that what's written now in section 4 does not add much on top of what I already understood from the intro. 3. The paper concludes by presenting an interesting set of experiments using both synthetic data and data that was obtained from tweeter. However, it seems to me that trimmed greedy obtains value very close to GSAS in tweeter data simulations. The paper must include some discussion on this issue. It would also help to compare to OPT, and to know the average number of rounds that GSAS ended up performing. Another two very minor comments are: (1) k was never explicitly defined as the cardinality parameter, (2) Algorithm 1 is randomized so it is confusing that Theorem 1 gives a deterministic statement – an explanation will be helpful I believe. Overall, in my opinion, studying GS functions is important, the adaptive complexity is a nice new measure, and the results (upper + lower bounds + experiments) could potentially tell a very nice, complete story. I hope that if the paper gets accepted the author(s) will clarify in the final version the above comments and questions as promised in the rebuttal.

Strengths: See above

Weaknesses: See above

Correctness: See above

Clarity: See above

Relation to Prior Work: YES

Reproducibility: Yes

Additional Feedback:

[Author Response · NeurIPS 2020]

We thank the reviewers for their insightful comments and suggestions. Reviewer-specific comments to follow.

**Reviewer 1.** Thank you for your enthusiastic review and for finding this work highly relevant to the NeurIPS community; we agree that gross-substitute functions are an important class of functions and that the adaptive complexity has practical use-cases using parallelization.

**Reviewer 2.** Thank you for your positive review and great suggestions. The question of GS + matroid is a very interesting direction for future work. Thank you for bringing this up and we agree this is missing in the current version of the manuscript and will address this in the next version. Regarding "the switch from submodular + matroid to GS + cardinality", the techniques in [7] for submodular + matroid consist of two parts, the adaptive sequencing technique together with a continuous greedy technique. The adaptive sequencing technique by itself is sufficient for submodular + cardinality but continuous techniques are required for matroid constraints. Here, for GS + cardinality, we modify the adaptive sequencing technique and do not use continuous techniques, which are very slow in practice.

Regarding "richer classes of gross substitute functions", we believe that the OXS valuations on graphs we consider are among the richest classes of GS functions. Indeed, to the best of our knowledge, all existing lower bounds for gross substitutes, including those in this paper, are constructed using OXS valuations, which implies that OXS valuations are among the hardest GS functions to optimize.

**Reviewer 3.** Thank you for your review. It seems like there is a simple misunderstanding about the algorithm which we discuss below. We hope that in light of our response you will consider revising your score.

- Regarding motivation for adaptivity: We would be happy to include a short paragraph summarizing the motivation and tie to the broad line of work on adaptivity discussed in related work.

- "what happens if, in line 8, the set $\{i$ s.t. $|X_i| < (1-\epsilon)|X|\}$ is empty": **See line 186, 187**: $i^\star$ is the largest position $i$ such that a large fraction of the elements in $X$ has high contribution to $S \cup A_{i-1}$. If $\{i$ s.t. $|X_i| < (1-\epsilon)|X|\} = \emptyset$, this implies that for all $i \in [k^\star]$, a large fraction of elements in $X$ have high contribution to $S \cup A_{i-1}$, in which case we add the entire sequence of elements $A_{k^\star}$ to the current solution $S$.

- "shouldn't $X$ in line 4 receive $N \setminus S$ and not $N$?": Both are correct. It is fine to have $X$ also receive the elements $a \in S$ as they have marginal contribution 0 to the current solution, i.e. $f_S(a) = 0$, which implies that these elements will all be removed from $X$ in the first iteration of the inner-while loop. To recap, after one iteration of the inner loop, $X$ will not contain elements from $S$ either way.

- "...$X_1$ will be the empty set. As a result line 9 will add a low value item to $S$." This is incorrect. If $X_1$ is the empty set, then we have $i^\star = 1$. **Line 9** states that $S \leftarrow S \cup \{a_1, \ldots, a_{i^\star-1}\}$. Thus, if $i^\star = 1$, we have $i^\star - 1 = 0$ and line 9 adds zero elements to $S$.

- "The paper does not really explain the algorithm." We refer the reviewer to **line 9**, which might add zero elements to $S$. We believe that the description of the algorithm clearly explains what happens in the case pointed out and that Reviewer 3 made a minor mistake when running the algorithm, which caused the confusion.

- "Section 4 gives some lower bounds but does not add really *any* new information on top of what the intro already told us.": The intro only informally states the lower bounds. Section 4 gives the precise statements of theorems for the lower bound. We believe that it is very important for every paper to contain precise statement of the main results.

- "it seems to me that trimmed greedy obtains value very close to GSAS in most of the figures." This is incorrect. In over half of the figures, GSAS outperforms TRIMMED-GREEDY by a factor of at least 2 on almost all values of $k$. (See **Figures 1a, 1b, 1c, 1d, and 3.**)

- "It would also help to compare to OPT": From **Theorem 3**, we know GSAS is arbitrarily close to $OPT$ and hence, they are empirically indistinguishable. We will clarify this in the next version of the manuscript.

- "the average number of rounds that GSAS ended up performing.": Thank you for the suggestion. GSAS uses significantly fewer rounds than $k$ in our experiments. We will include this in the next version of the manuscript.

- "$k$ was never explicitly defined as the cardinality parameter": Thank you, we will fix that.

- "it is confusing that Theorem 1 is gives a deterministic statement": Since the elements are chosen u.a.r. among all elements with high contribution, the guarantees hold deterministically.

- "is not polished enough to published": Please note that we worked very hard to make the paper polished and easy to follow. We note that both Reviewers 1 and 2 thought the paper is very well written. We believe that a main reason for this comment is the inablility to run the algorithm over an example due to a minor mistake in understanding. We hope that with the explanations and examples provided in this rebuttal the algorithm makes sense.

[Meta-Review · NeurIPS 2020]

The reviewers felt the technical contributions of the paper were strong, but that the presentation needs to be improved. In particular, the details of algorithm were not completely clear, and required knowledge of previous work to fully understand. We trust the authors will clarify the algorithm, and provide the explanations from the rebuttal in the paper itself.